# Enzymatic Conversion of Mannan-Rich Plant Waste Biomass into Prebiotic Mannooligosaccharides

**DOI:** 10.3390/foods10092010

**Published:** 2021-08-26

**Authors:** Nosipho Hlalukana, Mihle Magengelele, Samkelo Malgas, Brett Ivan Pletschke

**Affiliations:** Enzyme Science Programme (ESP), Department of Biochemistry and Microbiology, Rhodes University, Makhanda 6140, Eastern Cape, South Africa; g13h3281@campus.ru.ac.za (N.H.); g15m4462@campus.ru.ac.za (M.M.); b.pletschke@ru.ac.za (B.I.P.)

**Keywords:** endo-β-1,4-mannanase, mannan, mannooligosaccharide, gut microbiome, prebiotic, probiotic

## Abstract

A growing demand in novel food products for well-being and preventative medicine has attracted global attention on nutraceutical prebiotics. Various plant agro-processes produce large amounts of residual biomass considered “wastes”, which can potentially be used to produce nutraceutical prebiotics, such as manno-oligosaccharides (MOS). MOS can be produced from the degradation of mannan. Mannan has a main backbone consisting of β-1,4-linked mannose residues (which may be interspersed by glucose residues) with galactose substituents. Endo-β-1,4-mannanases cleave the mannan backbone at cleavage sites determined by the substitution pattern and thus give rise to different MOS products. These MOS products serve as prebiotics to stimulate various types of intestinal bacteria and cause them to produce fermentation products in different parts of the gastrointestinal tract which benefit the host. This article reviews recent advances in understanding the exploitation of plant residual biomass via the enzymatic production and characterization of MOS, and the influence of MOS on beneficial gut microbiota and their biological effects (i.e., immune modulation and lipidemic effects) as observed on human and animal health.

## 1. Introduction

Lignocellulosic biomass mainly consists of three polysaccharides (cellulose, hemicelluloses, and pectin) and an aromatic polymer (lignin) [1]. Hemicelluloses account for about 15 to 25% of all lignocellulose in dry mass basis [2]. In softwood species, legumes and agro-industrial by-products (spent coffee grounds and palm kernel press cake, among others), mannans are the most abundant hemicelluloses [3,4,5]. The huge amount of residual plant biomass considered as “waste” can potentially be used to produce various value-added products, such as prebiotics, biofuels, animal feeds, chemicals, enzymes, etc. [6]. It is this biomass which has the potential to be used for enzymatic generation of prebiotic mannooligosaccharides (MOS).

A complex dynamic relationship between the host and the gastrointestinal bacteria (microbiota) occurs shortly after birth [7]. The microbiota diversifies as a function of age to form an intestinal microbiota that is unique for each individual. Several findings suggest that the microbial cohort remains relatively constant once adulthood is reached. However, the composition of the resident microbiota may alter as a result of environmental factors, such as diet and antibiotic usage [8,9]. The gut microbiota intimately interacts with the host’s epithelial cells and stromal cells to play several crucial regulatory functions, such as regulating barrier function, maintaining mucosal immune homeostasis and host-microbiota symbiosis, prevention of pathogenic infections, and metabolism regulation [10]. As a result of these crucial roles exhibited by the microbiota, a healthy body is inseparable from an integrated gut to microbiota relationship [11]. Fostering the growth of these microbial species in humans is crucial in disease prevention and therapy. Hemicellulose-derived prebiotic oligosaccharides are able to selectively stimulate or inhibit the growth of specific microbial individual species, and this makes them ideal for applications in microbiota-linked disease prevention and therapy [12].

Simply defined, prebiotics are non-digestible food components that selectively stimulate the growth or activity of specific indigenous bacteria (probiotics), such as *lactobacilli* and *Bifidobacterium* and non-pathogenic yeast, *Saccharomyces boulardii*, while inhibiting the growth of toxin-producing bacteria, such as *Streptococcus pneumonia*, proteolytic *Clostridia* and *Escherichia coli*, in the digestive tract, in a manner claimed to be beneficial for the host [12,13,14]. Prebiotics are primarily carbohydrates (oligosaccharides and polysaccharides) in nature, with oligosaccharides included in this category being fructooligosaccharides (FOS), xylooligosaccharides (XOS), galactooligosaccharides (GOS), mannooligosaccharides (MOS) and pectin oligosaccharides (POS), as well as some sugar alcohols [15,16].

## 2. Mannan-Containing Waste Biomass Derived from Agro-Processing

Pineapples are tropical plants which belong to the family *Bromeliaceae* [17]. Pineapples are usually eaten as fresh fruit, but some pineapples are processed into fruit juice while others are canned [17,18]. The processing of pineapples for the food sector usually results in waste arising from the peel and pulp that is often left to rot on farms or is disposed of in ways that do not benefit the economy [19]. Research has been performed to find ways in which this waste stream can be beneficial to the economy. Pineapple waste has been shown to contain sucrose, as well as starch and hemicellulose [19,20,21]. These saccharides can be turned into value-added products that can be used to benefit the economy while also decreasing the waste produced and lessening the burden on landfills. One of the ways in which pineapple waste can be used as a value-added product is its use as an animal feed, particularly for monogastric animals [22]. However, the problem with using pineapple waste as an animal feed arises from its low protein content [23]. This is overcome by using sugars found in pineapple waste, especially those in the peel, for growing microorganisms that may be used to increase the protein content of the waste product [24]. This ensures that the animal feed industry is less reliant on traditional sources that have been used as protein for animal feeds [24]. Pineapple waste has also been used as a starting substrate to produce antioxidant products which offer health benefits [17]. Pineapples are among the most widely produced fruits in South Africa, with a total of 115,507 metric tons of pineapple production reported for the period 2018/2019 (https://www+.statista.com/statistics/1155961/production-of-pineapples-in-south-africa/, accessed on 16 August 2021).

The global forestry industry produces wood pulp and paper from softwood trees, such as pine, fir, and spruce trees. These softwoods are native to the Northern hemisphere countries, but it has been shown that they can also grow in the Southern hemisphere, however, they take a long time to grow in these regions, where they take anywhere from 20 to 60 years to fully mature [25]. The forestry industry produces a lot of waste, such as sawdust and bark. Biomass waste is projected to be about 2,100,000 metric tons in the 2017 to 2027 period, and this will be from pine and *Eucalyptus*. Data on the amount of pine waste currently produced in South Africa are difficult to obtain, but it is estimated that about 376,000 tons are produced annually [26]. Softwood derived sawdust hemicelluloses contain galactoglucomannan as the major hemicellulose, followed by arabinoglucuronoxylan, and arabinogalactan, and can be used for several applications in the bioeconomy, such as the production of bioethanol and use as functional foods [3]. The use of sawdust and wood shavings is important in the pharmaceutical industry since they have been reported to show antimicrobial, antioxidant, anti-inflammatory and anti-tumoral activities [27]. These bioactive properties are said to be due to the phenolic compounds in the bark of the pine tree. Sawdust and wood shavings from pine, therefore, have significant potential in the fields of food preservation and nutraceuticals [27].

Aloe plants, which consist of over 360 species known today, have long been known to have medicinal properties, especially in traditional medicine [28]. There is some debate as to which family the aloe plant comes from, as most of the literature suggests that it is part of the *Liliaceae* family, although it has also been said to belonging to a family of its own, *Aloaceae* [29,30]. Aloe plants are commonly found in dry regions of Africa, Asia, Southern Europe and the Mediterranean region [28]. The medicinal benefits of aloe plants are attributed to the entire leaf, the skin and the gel. The gel portion of the leaf is known to contain polysaccharides composed of monosaccharides such as mannose, glucose, arabinose and galactose [31]. Acetylated mannan, known as acemannan, constitutes approximately 50% of the polysaccharides found in *Aloe vera* gel, making it the most abundant polysaccharide in *A. vera* [32,33]. Studies on the differences in polysaccharide composition between *A. vera* and *A. arborescens* indicate that the different species contain the same types of polysaccharides, although they are found in different concentrations among the two species [31].

Coffee is the second largest traded commodity after petroleum. In 2014 and 2016, the International Coffee Organization (ICO) estimated that the amount of coffee consumed worldwide was 8.5 and 9.3 billion kg, respectively. Production of coffee results in the generation of spent coffee ground (SCG) which contains lignocellulosic material [34]. South Africa, Angola, Kenya, Uganda and Ethiopia are the major producers of SCG in Africa. SCG is rich in sugars and contains about 45.3% of them in dry mass basis [35]. Galactomannan, responsible for the high viscosity of coffee extract, is the major polysaccharide in SCG that forms about 20–30% dry matter and is water-insoluble [36,37]. SCG can be used for generation of various value-added products, such as enzymes [38], MOS [39] and fermentable sugars [5,40].

Palm kernel cake (PKC) is an agro-industrial waste produced from the extraction of palm oil. The worldwide production volume of palm oil increases every year. It has been reported that in the periods 2015 to 2016 and 2018 to 2019, between 7 and 8.55 million metric tons of PKC, respectively, were produced worldwide (https://www.statista.com/statistics/613479/palm-kernel-oil-production-volume-worldwide/, accessed on 11 October 2020). Nigeria, Indonesia, Malaysia and Thailand are the main producers of palm oil in the world [41]. The Singapore-listed agri-business, Wilmar, owns palm oil estates in Uganda and West Africa where palm oil plants are planted on 6000 and 39,000 hectares of land (https://www.wilmar-international.com/, accessed on 11 October 2020). The Societe Financiere des Caoutchoucs (SOCFIN) group owns more than 51,000 hectares in Nigeria, Ivory Coast and Cameroon (https://www.socfin.com/en, accessed on 11 October 2020). The chemical composition of PKC differs based on the type of fruit and the method used for oil extraction. Enzymatic hydrolysis of PKC is important for MOS production since PKC is rich in crude fiber, which mainly consists of cellulose and hemicellulose, where hemicellulose is composed of 58% linear mannan [18,42]. Because PKC has a high content of crude fiber, PKC is mostly used in feeding ruminants and limited in the non-ruminant diets e.g., poultry. Table 1 below shows the amount of mannan composition found in different biomass wastes generated from various agro-processes.

The above listed mannan-containing waste streams are usually generated from agricultural processing activities embarked upon in various agricultural and food sectors in various African markets and may serve as rich substrates for the enzymatic generation of the nutraceutical prebiotics. The advantage in the exploration of using agricultural crop residues as starting materials for nutraceuticals production is that it paves a way for the manufacturers to substantially reduce the cost of raw materials, as well as solve their disposal problems [47].

## 3. Mannan

Mannans are naturally found in four different forms, namely linear mannan, glucomannan, galactomannan, and galactoglucomannan [48]. Linear mannan is composed of linear chains of D-mannose residues linked by β-1,4-glycosidic bonds. The resemblance of the linear mannan backbone to cellulose makes it water-insoluble like the latter [49]. Glucomannan, on the other hand, is composed of a combination of D-mannose and D-glucose residues linked by β-1,4-glycosidic bonds. The M: G (mannose: glucose) ratio in glucomannans ranges from around 1.5:1 to 4.2:1 [50]. Glucomannans are hydrophilic and, as a result, are highly soluble in water [49]. Galactomannans, on the other hand, are composed of linear chains of D-mannose residues that are substituted by galactose residues via α-1,6-glycosidic bonds. The D-mannose/D-galactose substitution ratio of galactomannans differs from one gum to another (with the following ratios for locust bean (4:1), tara gum (3:1), guar gum (2:1) and (1:1) in fenugreek gum) and is responsible for other functional properties as well such as viscosity and gel-forming ability of the gum [48,49]. Galactoglucomannans, the most complex of mannans, are composed of a glucomannan backbone with D-galactose substitutions. The mannose, glucose, and galactose residues in galactoglucomannan are reported to be in the molar ratio of 3:1:1. Galactoglucomannans can be acetylated at the C-2 and C-3 positions of mannose residues to various degrees, depending on the source of the polysaccharide [51]. Figure 1 below illustrates the basic structural elements found in four types of mannans which occur in nature.

## 4. Endo-β-1,4-Mannanases

The β-1,4-D-mannan mannohydrolase (called β-mannanase, EC 3.2.1.78) is an endo-enzyme which is responsible for the random cleavage of β-1,4-linked internal linkages of the mannan backbone to mainly produce mannobiose and mannotriose, and additional traces of higher oligosaccharides in some instances [52,53]. In the Carbohydrate-Active Enzyme database (CAZY) (https://www.cazy.org/, accessed on 11 August 2020), mannanases are allotted to glycosyl hydrolase (GH) clan A, which is characterized by a (β/α)_8_-barrel protein fold, under GH families 5, 26, 113 and 134.

Mannanases are often modular, and in addition to catalytic domains, they may also have carbohydrate binding modules (CBM). The role of CBMs is to localize the soluble enzyme to its target substrate, and in some cases, it is also suggested that the CBMs are able to disrupt the structural integrity of the polysaccharide matrix in biomass, making it more accessible to enzymatic hydrolysis [54,55].

Most mannanases (GH5, 26 and 113) show a double displacement mechanism with retention of an anomeric configuration. This mechanism involves the attack of a nucleophile at the anomeric center with a general acid-catalyzed displacement of the leaving group. This forms a covalent glycosyl-enzyme acylal intermediate. A general base catalyzed process takes place which involves water attacking the anomeric center of the intermediate, yielding the product and releasing the enzyme to its original state [56]. Only GH5 and 113 mannanases have been reported to exhibit transglycosylation activity, while no such reports have been recorded for GH26 mannanases [57,58].

## 5. Enzymatic Production of MOS from Agro-Industrial Biomass

There are several studies reporting on the hydrolysis of mannans from agro-industrial wastes using mannanases. About 50.7% of the total mannans from coffee waste were hydrolyzed, producing mannooligosaccharides, using a GH5 mannanase derived from *Talaromyces trachyspermus* B168 [59]. Also, *an Aspergillus niger* derived mannanase (Man1) provided the highest increments in soluble solids yield (17%) from SCG hydrolysis [5]. A recent study showed that *A. quadrilineatus* RSNK-1 derived mannanase was very effective in the degradation of copra meal (CM), yielding 39% MOS after 1 h of hydrolysis [60]. Similarly, a *Streptomyces* sp. BF 3.1 derived mannanase produced 3.83 mg/mL of reducing sugars, including mannobiose, mannotriose, mannotetraose, mannopentaose, and mannohexaose, during the hydrolysis of 10% CM for 5 h [44]. A total MOS yield of 34.8 g/100 g dry PKC, indicating 80.6% hydrolysis of total mannan in PKC, was achieved using a GH5 mannanase derived from *Rhizomucor miehei* [61]. A study on the hydrolysis of spruce derived galactoglucomannan by two mannanases from *Cellvibrio japonicus* showed that CjMan5A action resulted in higher amounts of mannotriose and mannotetraose than that of CjMan26A, which mainly generated mannose and mannobiose as end products [62].

## 6. Identification and Quantification of Mannanase Produced MOS

Produced MOS can be identified using a variety of methods which include chromatography and electrophoretic gels. Thin layer chromatography (TLC) is one of the most used methods to determine the degree of polymerization (DP) of produced MOS [63]. TLC uses coated plates as the stationary chromatographic phase for the separation of oligosaccharides of different DP. TLC is a rapid and cheap method of analyzing oligosaccharides [64]. This method uses silica gel, aluminum, or cellulose plates as the stationary phase and uses a solvent as the mobile phase for separation of the MOS. The hydrolysis products are visually observed as spots on the silica plate that have migrated differently based on their size, with the low DP MOS moving along faster and further up the plate. Using TLC also allows for multiple samples to be viewed on one plate for a better comparison of various samples [64]. The use of TLC has been adapted to more sophisticated techniques that allow for better quantification of oligosaccharides. These include high performance-TLC (HPTLC) which is a rapid and cost-effective method that offers a high sensitivity of analysis and produces reproducible results [65]. The use of HPTLC also requires the use of a small amount of sample, which means that MOS produced will not be wasted during their analysis [65]. HPTLC works on the same principle as normal TLC, having a solvent mobile phase that moves along the stationary phase due to capillary action, which helps the analytes that have a lower affinity for the stationary phase to move faster along the plate than those with a higher stationary phase affinity which move slower along with the plate.

Fluorophore-assisted carbohydrate electrophoresis (FACE) is another method that can be used for the visualization of MOS. In FACE, a fluorescent dye is attached to the reducing end of the oligosaccharide and is followed by high resolution viewing on a polyacrylamide slab gel [66]. Polyacrylamide gels require negatively charged molecules to move through the electric field, which is not usually the case for most carbohydrates [66]. This means that carbohydrates that are not charged or those which have a net positive charge need to be tagged with a fluorophore that will give them a net negative charge in order for them to be able to move on the FACE gel [66,67]. Some fluorophores used to tag these carbohydrates are 8-amino-1,3,6-naphthalenetrisulfonic acid (ANTS) and 8-amino-1,3,6-pyrenetrisulfonic acid (APTS) [67]. The oligosaccharides migrate through the gel based on size, and the abundance of the oligosaccharide being analyzed is measured by the intensity of the band that is represented by it [68].

Structural analysis of MOS can be achieved by using spectroscopic methods, including mass spectrometry (MS), Fourier-transform infrared spectroscopy (FTIR) and 1D/2D nuclear magnetic resonance (NMR) [32,33,69]. The structural analysis of MOS can provide more information than that offered by visualization techniques such as FACE and TLC (limited to DP analysis), it offers insights on glycosidic linkages in the MOS (using MS or NMR), the degree and extent of substitution of MOS by substituents such as galactosyl substituents and acetyl groups (using NMR or FTIR), and in some cases the molecular masses of the MOS (using MS) [69,70,71]. Using 2D NMR, data from ^13^C NMR can be used to show the presence of a glycosidic bond (β-1,4) by the resonance of C4 in reducing D-mannose at 81.1 ppm, while from ^1^H NMR spectra the resonances at 5.2 ppm (MRα) and 4.7 ppm (MNRβ) can be assigned to the anomeric protons of the reducing and non-reducing termini of the MOS, respectively [69].

## 7. MOS Generation Using Mannanases from Various GH Families

To date, it is predominantly mannanases from families GH5 and GH26 that have been used in studies to investigate the production of different types of MOS or the analysis of mannanase hydrolysis products patterns using model mannan substrates and/or mannan-containing agricultural residues (Table 2). The yield and product profile of MOS from the hydrolysis of mannans by mannanases varies according to the source or type of the mannan hydrolyzed, the enzyme activity or specificity as well as incubation conditions such as pH, hydrolysis time, the temperature of incubation, etc. Regarding processing conditions (e.g., temperature, pH and reaction time), bacterial mannanases appeared to be more alkalophilic (pH 6.0 to 8.0), while fungal mannanases were acidophilic (pH 4.0 to 6.0) (Table 2). Generally, mannanases display optimal activity at a temperature range of 30 to 50 °C and about 12 to 48 h to produce high quantities of MOS (Table 2). During MOS production, mannanases should ideally have no or lower exo-mannanase (β-mannosidase) activity so that mannose production can be minimized in the MOS mixture.

The most common MOS produced by enzymatic hydrolysis from mannans seem to be mannobiose (M2), mannotriose (M3) and mannotetraose (M4) (Table 2). Notably, GH5 mannanases did not produce higher DP MOS compared to those produced by GH26 mannanases even though this family of mannanases is reported to display transglycosylation activity. Subtle differences in MOS production have been observed in isoforms (same GH family) of the same enzyme from different species, for example, Man5HJ14 from *Bacillus* sp. HJ14 releases M1-M7 from LBG, while Bpman5 from *Bacillus pumilus* CBSW19 releases M1–M3, M5, M6 from the same substrate. Secondly, differences in MOS production have been observed in mannanolytic enzymes from different GH families and EC classifications from the same organism, for example, *C. japonicas* derived CjMan5A, CjMan26A and CjMan26C release different products from LBG, M1–M3; M1–M4; and M2, GM2, respectively. Also, *Bacteroides ovatus* derived BoMan26A and BoMan26B also release different products from LBG hydrolysis, M2, GM3; and M1–M6, respectively.

## 8. Properties and Biological Action of MOS

As described previously, MOS are short-chain carbohydrates responsible for enhancing the growth of beneficial bacteria while inhibiting the growth of pathogenic bacteria in the digestive tract of mammals. Chacher et al. [95] reported that MOS achieve this effect by binding to the type-1 fimbriae of gram-negative bacteria. Type-1 fimbriae contain mannose-specific lectins which have a high affinity for mannose residues. Binding of MOS to gram-negative bacteria then prevents these bacteria from attaching to the intestinal mucosa. This causes these bacteria to pass through the intestines, leading to a decrease in these gram-negative bacteria. In the intestine, MOS also increase the production of goblet cells situated in the membrane of the villi [95,96]. Goblet cells are well known for secreting mucus, which protects the mucous membranes. These cells also produce glycoproteins called mucins. Mucins contain mannosyl receptors and use these receptors to bind to type 1 fimbriae of gram-negative bacteria, assisting in the decrease of these bacteria in the digestive tract [95,96].

The utilization of MOS by beneficial bacteria such as *Lactobacillus* sp. and *Bifidobacteria* sp. results in the production of short-chain fatty acids (SCFA) and lactic acid (LA), which results in the decrease of the pH in the intestine preventing the attachment of pathogens such as *Escherichia coli* and *Clostridium perfringens* to the intestinal mucosa [39,95]. The decrease of these gram-negative bacteria then favors the growth of beneficial bacteria such as *Lactobacillus* sp. and *Bifidobacteria* sp. Asano et al. [39] demonstrated that the intake of 1 g/day and 3 g/day of MOS increases the content of *Bifidobacterium*. The content was slightly different, with more bacterium content in 3 g/day but the difference was not significant. Recently, it has been shown that mannobiose and mannotriose are the preferred oligosaccharides among the probiotic *Lactobacillus* sp., rather than functional oligosaccharides with higher degrees of polymerization [80]. As shown in Figure 2 below, the decrease of pathogens and the increase of beneficial bacteria results in the decrease of crypt depth and the increase in the height of the villi.

Intake of MOS triggers an immune response by using pathogen-associated molecular patterns (PAMP) to bind to the PAMP receptors of gut-associated lymphoid tissue (GALT) macrophages [95]. Oligosaccharides with a DP that is greater than six have been reported to be effective at influencing the immune response [97]. MOS cause an increase in the plasma levels of immunoglobulin G (IgG) and IgA in poultry [98]. MOS may probably have the ability to elicit powerful antigenic properties. The intake of MOS also results in the production of the mannose-binding protein by the liver. This protein plays an important role in first-line host defense [98]. Mannose-binding proteins move through the plasma as complexes attached to mannan-binding lectin serine protease 1 (MASP-1) and MASP-2 [99]. These complexes bind to the type-1 fimbriae situated on the surfaces of pathogens. MASPs are also involved in the lectin pathway of the complement system whereby they are responsible for the cleavage of complement protein C4 and C2 into fragments to form C3-convertase which is responsible for eventuating opsonization of particles, the release of inflammatory peptides and lysis of dying host cells [99].

Antioxidants are required to prevent and delay the binding of oxidants to cellular molecules such as DNA and proteins leading to adverse effects on the body such as cardiovascular diseases, liver diseases, diabetes, ageing, skin diseases, etc. [100]. Antioxidants achieve this by donating a hydroxyl group to the radical group on oxidants. The antioxidant activity of MOS can be determined by their ability to scavenge free radicals in their environment [101]. The presence of free radicals is associated with an increase in pathogenicity of certain diseases, caused by increased oxidative stress on the body, therefore, MOS with antioxidant ability would be beneficial in overcoming the effects that these diseases have on the host [101,102]. Lipid oxidation in foods can also lead to rapid food spoilage, which would result in the food having undesirable flavors and odors [103]. Studies conducted on the antioxidant activity of MOS include their ability to scavenge 2,2-diphenyl-1-picrylhydrazyl (DPPH), 2,2′-azinobis (3-ethylbenzothiazoline-6-sulfonic acid) (ABTS+•) and hydroxyl radicals [90,102]. Jana and Kango [90] conducted radical scavenging assays for MOS generated from mannan substrates with different substitution patterns which resulted in different MOS profiles. Their study found that palm kennel cake had more ABTS+• radical scavenging ability compared to other MOS evaluated, but there was a general stable scavenging ability across all the evaluated MOS for the DPPH.

Lipidemic effects of oligosaccharides have been evaluated for fructooligosaccharides (FOS) and some inulin prebiotics [104,105]. These studies have found that supplementing diets with prebiotics had an influence on lipid metabolism where rats showed a decrease in cholesterol levels after being fed FOS [104]. A proposed lipid-lowering mechanism is that fermentation of prebiotics in the gut results in increased fermentation products which then trigger hepatic cholesterol synthesis. It is likely that the fermentation of MOS in the gut may also result in these fermentation products which may trigger hepatic cholesterol synthesis which would, in turn, affect lipid metabolism.

Lee and co-workers reported that coffee (which is generally rich in mannans) consumption, especially by men, has the ability to reduce obesity [106]. Reduction of obesity as a result of coffee consumption is suspected to be due to the presence of bioactive compounds, such as chlorogenic acid, trigonelline, and caffeine [106]. Obesity is a serious health problem that can lead to diseases such as heart disease and diabetes. Many other studies have also reported that coffee also decreases the risk of type 2 diabetes and cardiovascular diseases [107,108]. Coffee extract treatment of adipose cells, 3T3-L1, showed that coffee disrupts their cell cycle by preventing it from continuing into the G2/M phase [107]. Coffee also results in the degradation of the molecule involved in insulin signaling, insulin receptor substrate 1 (IRS1) [107]. The molecular mechanism of how coffee intake reduces the risk of different diseases is still unclear, because previous reports are observational [107].

A few studies have also implicated MOS in eliciting anti-cancer bioactivity. MOS obtained from palm kernel cake and copra meal reduced the viability of a human colon cancer cell line, Caco-2, by 74.19%, and 62.21%, respectively [90]. Ghosh and co-workers [109] also demonstrated that MOS obtained from copra meal hydrolysis have the ability to reduce survival of the human colon cancer cell line, HT-29, by 50%, while guar gum reduced Caco-2 cell line survival by 30.95%.

Recently, a study showed that a yogurt containing a symbiotic concoction composed of konjac glucomannan derived oligosaccharides (KMOS) and *Bifidobacterium animalis* ssp. *lactis* BB12 (BB12) improved black fecal weight and number and gastrointestinal transit rate of diphenoxylate-induced constipated Kunming mice [110]. Additionally, restoration of intestinal damage, secretion of excitability neurotransmitters, and expression of components of the SCF/c-Kit pathway were also notably enhanced in mice administered with yogurt containing KMOS and/or BB12 [110].

SCFAs, such as acetate, generated during prebiotics fermentation by microorganisms may have a detrimental effect on the lipid profile [107]. Recommended prebiotic dosage is between 1.5 to 5 g/day, overdose of prebiotics may lead to side effects, such as osmotic diarrhoea and flatulence [107,108]. Rapid fermentation of carbohydrates also leads to intestinal bloating and excessive gas formation, leading to abdominal pain [109]. The adverse and beneficial effects of prebiotics vary between individuals and are influenced by intestinal oligosaccharide fermentation [109].

## 9. Conclusions

This review highlights recent advances made in enzyme technology for the production of prebiotic MOS from mannan-containing lignocellulosic wastes generated from various agro-processes applied in the African agricultural sector. The structural diversity of mannanase produced MOS depends on the type of substrate, pretreatment and structure/function of the mannanase utilized. Most of the mannanases utilized for MOS production to date belong to GH5 and GH26, GH26 being more tolerant in utilizing the decorated galactomannans as a substrate, while the GH5 mannanases are more restrictive in binding decorated mannans in their active sites and may also exhibit undesirable transglycosylation activity during MOS production alongside GH113 mannanases. A growing body of knowledge on the diverse specificity of various GH family mannanases at a structural level and utilization of mannans derived from different agricultural residues is anticipated to bolster improvements in the yields of MOS produced, as well as in a higher diversity in the MOS produced.

Further investigations are required to deeply elucidate the mechanisms involved in the biological effects demonstrated by MOS, particularly their immune modulation and lipidemic effects on humans and animals. It still remains elusive as to which is the ideal composition of the gut microbiota and whether MOS can induce the ideal range of gut microbiota. It’s also still unclear as to what the appropriate dose of these dietary modulators would be for improving health. In addition, the knowledge on the safety of prebiotics is still in its infancy, over time they may very well contribute to antibiotic resistance and prevent gastrointestinal disorders.

## Figures and Tables

**Figure 1 foods-10-02010-f001:**
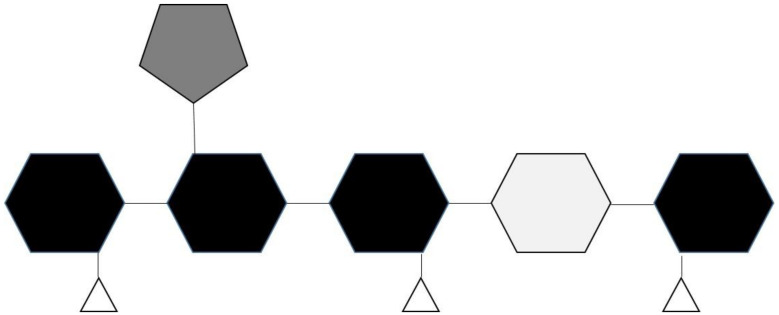
The basic structural components found in mannans. Their backbone is made of β-1,4-linked mannose residues “black hexagon” which may be randomly interspersed with glucose residues “light grey hexagon”, the mannose residues may also contain α-1,6-linked galactose sidechains “dark grey pentagon”, and acetyl group substitutions at the O-2 and O-3 positions “white triangles”.

**Figure 2 foods-10-02010-f002:**
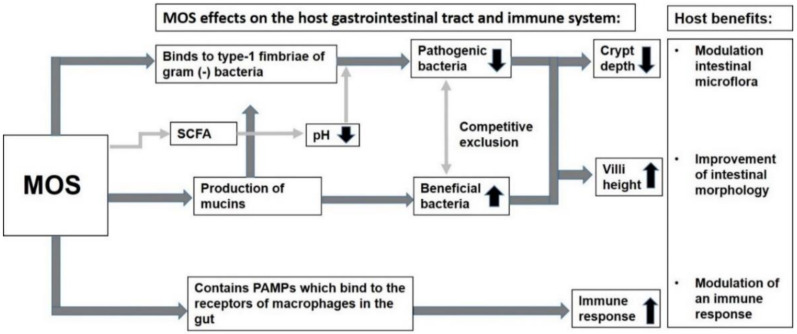
Mechanism of action of MOS. The diagram shows how MOS modulate the intestinal microflora, improve the intestinal morphology and elicit an immune response. The black arrows pointing upwards mean an “increase” in response, while the arrows pointing downwards mean a “decrease” in response.

**Table 1 foods-10-02010-t001:** The amount of mannan present in different biomass wastes.

Biomass Waste	Mannan Composition (%), Dry Mass Basis	Type of Mannan	References
SCG	20	Galactomannan	[5,34,40]
PKC	60	Linear mannan	[43]
CM	61	Linear mannan	[44]
PP	1.5	Galactomannan	[21]
PSD (softwoods)	12	Acetyl-galactoglucomannan	[45,46]
*A. vera*	50	Acetyl-galactomannan	[32]

SCG-spent coffee grounds, PKC-palm kernel cake, CM-copra meal, PP-pineapple pulp, PSD-pine sawdust.

**Table 2 foods-10-02010-t002:** Production of MOS by mannanases from various mannan and mannan-containing agricultural residues.

Enzyme Name and Source	Mannan Substrate	Parameters (pH, °C, Hours)	MOS Produced	Analytical Technique	References
GH5 mannanases					
CjMan5A, *Cellvibrio japonicas*	GGM KG LBG	7, 35, 24 h	GGM: M1–M4 KG: M1–M4 LBG: M1–M3	HPAEC-PAD: Carbopac PA200 column	[62]
PaMan5A, *Podospora anserine*	INM KG LBG	5.2, 40, 0.5 h	M1–M3	HPAEC-PAD: Carbo-PacPA-1 column	[72]
ManAK, *Aspergillus kawachii* IFO 4308	LBG KG GG	3.0, 60, 12 h	LBG: M1–M4, M6 KG: M1–M6 GG: M1–M4	TLC, HPLC: TSKgel PWXL column	[73]
HhMan, *Hypothenemis hampei*	LBG GG	5.5, 30, 24 h	M1–M4	TLC	[74]
BlMan5_8, *Bifidobacterium animalis*	INM LBG	6.0, 37, 0.0083 h (30 s)	INM: M3–M5 LBG: M2–M5	HPAEC-PAD: CarboPac PA200 column	[71]
mRmMan5A, *Rhizomucor miechei*	PKC	4.5, 50, 8 h	M1–M4	HPLC-ELSD, Sugar KS-802 column	[61]
TtMan5A, *Talaromyces trachyspermus* B168	DCW	4.5, 50, 69 h	M1–M4	HPAEC-PAD DX-500	[59]
Bpman5, *Bacillus pumilus* CBSW19	LBG	6.5, 50, 24 h	M1–M3, M5, M6	TLC, HPLC: C18 reverse phase column	[75]
ManF3, *Aspergillus niger* BCC4525	CM PKC	5.5, 50, 24 h	CM: M1–M5 PKC: M1–M3	TLC	[76]
	Mannan	5.5, 50, 0.17–3 h	0.17 h: M2–M5 3 h: M2–M7	HPAEC-PAD	[77]
ManPN11, *Bacillus nealsonii* PN-11	LBG	8.8, 55, 6 h	M1–M5, >M5	HPAEC: CarboPac PA100 column	[78]
ManPN11, *Bacillus nealsonii* PN-11	CB	8.8, 37, 24 h	M1–M5	TLC, HPLC: Aminex HPX-87P column	[79]
Man5HJ14, *Bacillus* sp. HJ14	LBG PKC	6.5, 60, 10 h	LBG: M1–M7 PKC: M1–M4	TLC, ESI-MS	[80]
GH26 mannanases					
PaMan26A, *Podospora anserine*	INM KG LBG	5.2, 40, 0.5 h	M1–M4	HPAEC-PAD: Carbo-PacPA-1 column	[72]
BoMan26B, *Bacteroides ovatus*	GG LBG	6.5, 37, 24 h	GG: M1–M6 LBG: M1–M6	HPAEC-PAD: CarboPac PA200 and PA20 columns	[81]
CjMan26A, *Cellvibrio japonicas*	GGM KG LBG	7.0, 37, 24 h	GGM: M1–M4 KG: M1–M3 LBG: M1, M2	HPAEC-PAD: CarboPac PA200 column	[62]
Mannanase, *Bacillus circulans*	LBG	6.0, 50, 24 h	M2–M6	HPLC: Aminex-HPX42C column	[82]
BoMan26B, *Bacteroides ovatus*	GG INM LBG	6.5, 37, 24 h	GG: M2–M6, >M6 INM: M2 LBG: M2–M6, >M6	HPAEC-PAD: CarboPac PA200 column	[83]
ManAJB13, *Sphingomonas* sp. JB13	LBG PKC	6.5, 37, 10 h	LBG: M1–M6 PKC: M1–M4	TLC, ESI-MS	[80]
GH26 mannobiohydrolases					
BoMan26A, *Bacteroides ovatus*	GG INM LBG	6.5, 37, 24 h	GG: M2, GM3, M5 INM: M2 LBG: M2, GM3	HPAEC-PAD: CarboPac PA200 column	[83]
CjMan26C, *Cellvibrio japonicas*	INM LBG	37, 7.0, 0.5 h	INM: M2 LBG: M2, GM2	HPLC: CarboPac PA100 column	[84]
RSMan26H, *Reticulitermes speratus*	INM	5.5, 30, 1 h	M2–M5	HPLC: Shodex Asahipak NH2P-50 4E column	[85]
GH113 mannanases					
AaManA, *Alicyclobacillus acidocaldarius*	KG LBG	5.5, 65, 3 h	KG: M1–M6 LBG: M1–M6, >M6	TLC	[86]
BaMan113A, *Bacillus* sp. N16- 5	KG LBG	7.0, 30, 2 h	KG: M1 LBG: G, GM, GM2, M1, M2, M4	TLC	[87]
GH134 mannanases					
AnMan134A, *Aspergillus nidulans*	INM KG LBG	6.0, 37, 0.25 h	INM: M2–M4 KG: M2–M6, >M6 LBG: M2–M6, >M6	TLC, MALDI-TOF-MS, HPLC: Shim-pack ISA-07/S2504 column	[88]
Unidentified mannanases					
Mannanase, *Bacillus* sp. GA2(1)	SCG	6.0, 50, 5 h	M2, M3	TLC	[34]
ManAo, *Aspergillus oryzae*	GG KG LBG	5.0, 60, 24 h	GG: M1–M4, >M4 KG: M1, M2 LBG: M1–M3, >M4	FACE	[89]
ManAo, *Aspergillus oryzae*	CM GG KG LBG PKC	5.0, 50, 12 h	CM: M1–M3 GG: M1–M3 KG: M1–M4 LBG: M1–M3 PKC: M1–M3	HPLC: Sugar-Pak column, 2D NMR	[90]
rHhMan, *Hypothenemus hampei*	GG LBG	5.5, 30, 24 h	M1–M4	TLC	[74]
Mannanase, *Streptomyces cyaenus*	PKC	6.0, 37, 8 h	M2–M6	TLC, HPLC: Zorbax carbohydrate column	[91]
Mannanase, *Kitasatospora* sp. KY576672	SPF	6.0, 40, 32 h	M1–M6	TLC, HPLC: Hi-PlexCa (Duo) column	[92]
Mannanase, *Kitasatospora* sp.	INM LBG PP SPF	6.0, 40, 2–6 h	INM: M1–M5 LBG: M1–M6, >M6 PP: M1–M6, >M6 SPF: M1, M2, >M6	TLC	[93]
Mannanase, *Penicillium aculeatum* AP S1	GG KG LBG	5.3, 50, 3 h	GG: M1–M3, >M4 KG & LBG: M1–M4, >M4	TLC, HPLC: Hamilton RCX-30 column	[94]

Substrates: CB-coffee beans, CM-copra meal, DCW-dried coffee waste, GG-guar gum, GGM-galactoglucomannan, INM-ivory nut mannan, KG-konjac glucomannan, LBG-locust bean gum, PKC-palm kernel cake, PKM-palm kernel meal, PP-porang potato, SCG-spent coffee grounds, SPF-sugar palm fruit. Standards: G-galactose, GM-galactosyl-mannose, GM2-galactosyl-mannobiose, GM3-galactosyl-mannotriose, M1-mannose, M2-mannobiose, M3-mannotriose, M4-mannotetraose, M5-mannopentaose, M6-mannohexaose. Analytical techniques: FACE-fluorescence assisted carbohydrate electrophoresis, ESI-MS-electrospray ionization mass spectrometry, HPLC-high-performance liquid chromatography, HPAEC-PAD-high-performance anion-exchange chromatography with pulsed amperometric detection, MALDI-TOF-MS-matrix-assisted laser desorption/ionization time-of-flight mass spectrometry, NMR-nuclear magnetic resonance spectroscopy, TLC-thin layer chromatography.

## Data Availability

No new data were created. Data sharing not applicable.

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
