# Peer review of "Enzymatic Conversion of Mannan-Rich Plant Waste Biomass into Prebiotic Mannooligosaccharides"

_foods, 2021, doi:10.3390/foods10092010_

Round 1

Reviewer 1 Report

The authors Hlalukana et al. have critically reviewed the role of Mannan-rich plant conversion to a prebiotic mannooligosaccharides. The paper is very-well written, clear in describing the important aspects of the process. 

  1. We know the fibers are beneficial for health, however, not all prebiotics are good in every disease/health states. It would be best to add a comment on that, basing on current literature.
  2. Safety would well be dependent on the dose/dosing strategy. A comment on it will be beneficial. 

Author Response

I have no major objection to the manuscript, but I have minor:

  1. I propose to make the Table 1 horizontally - it will be more transparent

The table has been landscaped as suggested by the reviewer (and has also been changed to Table 2)

  1. For Authors to consider - make a list of abbreviations that are abundant in the manuscript

A list of abbreviations has been added as suggested by the reviewer, however, it is unclear where to place this on the paper, as the manuscript template has no provision for abbreviations.

Listed below is the list of compiled abbreviations:

ABTS                     2,2'-azinobis (3-ethylbenzothiazoline-6-sulfonic acid)

DP                         Degree of polymerization

DPPH                    2,2-diphenyl-1-picrylhydrazyl

EC                         Enzyme commission number

FOS                       Fructooligosaccharide(s)

GH                        Glycoside hydrolase

FACE                     Fluorophore-assisted carbohydrate electrophoresis

GG                        Guar gum

GOS                      Galactooligosaccharide(s)

HPLC                    High-performance liquid chromatography

PKC                       Palm kernel cake

RID                        Refractive index detector

SCFA                     Short-chain fatty acids

INM                      Ivory nut mannan

LBG                       Locust bean gum

MOS                     Mannooligosaccharide(s)

MS                        Mass spectrometry

NMR                     Nuclear Magnetic Resonance

SCG                       Spent coffee grounds

SCFA                     Short-chain fatty acid(s)

TLC                       Thin-layer chromatography

XOS                       Xylooligosaccharide(s)

  1. Some minor errors were noted in the submitted manuscript (line 76, 96-98, 109, 132, 383, 387, 388, 446, 459, 489)

“sp” has been changed to non-Italic as suggested by the reviewer

76- Bromeliaceae written in italics

132- A. vera gel written in full

383- Bifidobacterium written in italics

387-Lactobacillus written in italics

388- Figure 3.2 corrected to figure 2

488 and 489- British to American English

Reviewer 2 Report

The topic of the article falls within the thematic scope of the journal.

The manuscript reviews the current knowledge (approximately 50% of the cited literature sources were published in the last 5 years) on the use of mannan-rich plant waste products (mainly available in Africa, but not limited to) for enzymatic conversion into products that could be used as prebiotics. 

I have no major objection to the manuscript, but I have minor:

1. I propose to make the Table 1 horizontally - it will be more transparent 

2. For Authors to consider - make a list of abbreviations that are abundant in the manuscript 

3. Some minor errors were noted in the submitted manuscript (line 76, 96-98, 109, 132, 383, 387, 388, 446, 459, 489).

Author Response

The topic of using waste from agricultural and food production is topical and very important all over the world. It was also undertaken by the Authors of the reviewed article. I have some comments and remarks on the reviewed manuscript:

  • line 69: "fructo-oligosaccharides" please write without the dash (fructooligosaccharides)

The dash has been removed as suggested by the reviewer

  • lines 75-76: "Bromeliaceae" please write in italics

"Bromeliaceae" has been written in italics as suggested by the reviewer

  • line 98: are there newer data than for period 2011/2012 on the production of pineapples in South Africa?

2018/2019 data on production of pineapples is added as requested by the reviewer, see lines 95-99.

  • lines 123-124: "Liliaceae" please write in italics

"Liliaceae" has been written in italics as suggested by the reviewer

  • line 125: "Aloaceae" please write in italics

"Aloaceae" has been written in italics as suggested by the reviewer

  • line 132: "A. vera gel" please write "Aloe vera gel"

Aloe vera gel has been written in full as suggested by the reviewer

  • lines 147-149: sentence unnecessary - it is not directly related to the use of post-production waste to obtain prebiotics

The sentence in line 147-149 has been removed

  • from the description in lines 137-149 it does not appear whether sugars are obtained from SCG or if it is just an idea for obtaining them - more and more accurate information is needed

This has been clarified, see lines 147-149.

  • line 383: "Bifidobacterium" please write in italics

Bifidobacterium written in italics as suggested by the reviewer

  • line 387: "Lactobacillus" please write in italics

"Lactobacillus" and "Bifidobacterium" in line 383 and line 387 have been written in italics as suggested by the reviewer

  • lines 446-454: does the described effect of coffee on the human body result from the properties of mannans or other / also other ingredients? More and more accurate information is needed.

Possible compounds responsible for the described effect of coffee on the human body have been alluded to, see lines 447-449 and 456-458.

Reviewer 3 Report

The topic of using waste from agricultural and food production is topical and very important all over the world. It was also undertaken by the Authors of the reviewed article. I have some comments and remarks on the reviewed manuscript:

  • line 69: "fructo-oligosaccharides" please write without the dash (fructooligosaccharides)
  • lines 75-76: "Bromeliaceae" please write in italics
  • line 98: are there newer data than for period 2011/2012 on the production of pineapples in South Africa?
  • lines 123-124: "Liliaceae" please write in italics
  • line 125: "Aloaceae" please write in italics
  • line 132: "A. vera gel" please write "Aloe vera gel"
  • lines 147-149: sentence unnecessary - it is not directly related to the use of post-production waste to obtain prebiotics
  • from the description in lines 137-149 it does not appear whether sugars are obtained from SCG or if it is just an idea for obtaining them - more and more accurate information is needed
  • line 383: "Bifidobacterium" please write in italics
  • line 387: "Lactobacillus" please write in italics
  • lines 446-454: does the described effect of coffee on the human body result from the properties of mannans or other / also other ingredients? More and more accurate information is needed.

Author Response

Few suggestions to further improve the quality of this manuscript is given below;

Scientific names should be in Italic throughout the manuscript. For example, in Line 63 – Bifidobacterium should be in Italic. Please correct throughout the manuscript.

Scientific names of organisms have been Italicised as suggested by the reviewer.

  1. Mannan-containing waste biomass derived from agro-processing in Africa – In this section you have discussed agro processing by-products around the globe, not only in Africa. This sub-title may need to change slightly reflect that.

Title changed from “Mannan-containing waste biomass derived from agro-processing in Africa” to “Mannan-containing waste biomass derived from agro-processing” as suggested by the reviewer, see section 2.

I also would like to suggest including a brief table explaining the summary of mannan-containing waste biomass derived from agro-processing in the world. This will further increase the value of this manuscript. I believe that there are many other relevant mannan-containing waste biomass derived from agro-processing industries and including these will be highly valuable for the readership.

Mannan-containing substrates are included in Table 1 along with their mannan types and percentage composition.

Lines 375-376 – Lactobacillus sp and Bifidobacterium sp – in here “sp” should not be in Italic. Please check carefully and correct. I can see you have used Italic font for “sp” throughout the manuscript. Please correct.

“sp” has been changed to non-Italic as suggested by the reviewer.

Is there any health hazards associated with the consumption of Prebiotic Mannooligosaccharides/ Mannan-rich Plant Waste Biomass? If so please mention those in this manuscript.

Hazards associated with the consumption of oligosaccharide-based prebiotics have been cited, see lines 475-483.

Please read the whole manuscript one more time to minimise the minor typos and associated errors

Manuscript has been proof read as suggested.

Reviewer 4 Report

This review on Enzymatic Conversion of Mannan-rich Plant Waste Biomass into Prebiotic Mannooligosaccharides by Hlalukana et al is an interesting article and on a timely topic. There is an increasing interest on gut microbiome and health and associated food components probiotics and prebiotics. Conversion of food wastes int edible materials or extraction of valuable components from such materials is also highly attractive research area at the moment. Hence this review provides a valuable insight in t many different crucial aspects in food and pharmaceutical sciences.

This manuscript is generally well written as well. The use of relevant references seems appropriate and relevant and current papers have been included in here.

Few suggestions to further improve the quality of this manuscript is given below;

Scientific names should be in Italic throughout the manuscript. For example, in Line 63 – Bifidobacterium should be in Italic. Please correct throughout the manuscript.

  1. Mannan-containing waste biomass derived from agro-processing in Africa – In this section you have discussed agro processing by-products around the globe, not only in Africa. This sub-title may need to change slightly reflect that.

I also would like to suggest including a brief table explaining the summary of mannan-containing waste biomass derived from agro-processing in the world. This will further increase the value of this manuscript. I believe that there are many other relevant mannan-containing waste biomass derived from agro-processing industries and including these will be highly valuable for the readership.

Lines 375-376 – Lactobacillus sp and Bifidobacterium sp – in here “sp” should not be in Italic. Please check carefully and correct. I can see you have used Italic font for  “sp” throughout the manuscript. Please correct.

Is there any health hazards associated with the consumption of Prebiotic Mannooligosaccharides/ Mannan-rich Plant Waste Biomass? If so please mention those in this manuscript.

Please read the whole manuscript one more time to minimise the minor typos and associated errors.

Author Response

(The authors gave the same response as above.)
